# Contribution of International Projects to the Development of Maritime Spatial Planning Structural Elements in the Northern Adriatic: The Experience of Slovenia

Gregor Čok [1,*], Slavko Mezek [2], Vane Urh [3] and Blaž Repe [4]

1 Faculty of Civil and Geodetic Engineering, University of Ljubljana, Zoisova 12, 1000 Ljubljana, Slovenia
2 Regional Development Centre Koper, Ulica 15. maja 19, 6000 Koper, Slovenia; slavko.mezek@rrc-kp.si
3 Development Centre Novo Mesto, Podbreznik 15, 8000 Novo Mesto, Slovenia; vane.urh@rc-nm.si
4 Faculty of Arts, University of Ljubljana, Aškerčeva 2, 1000 Ljubljana, Slovenia; blaz.repe@ff.uni-lj.si
* Correspondence: gregor.cok@fgg.uni-lj.si; Tel.: +386-1476-8645

**Abstract:** Maritime spatial planning (MSP) has been developing for years on the basis of international commitments, national legislations, and professional practices. Projects under European Territorial Cooperation have also made an important contribution to its development. They were designed to support EU countries in the implementation of MSP. The projects implemented in Slovenia always covered the entire national sea and coastal zone. In accordance with the MSP Directive, the countries of Northern Adriatic are currently preparing the first generation of maritime spatial plans, largely based on the experience and results gained from these projects. This article presents the results of research aimed at assessing the contribution of the projects to the preparation of the first plan in Slovenia. Using a descriptive research method, a detailed analysis of the results of seven projects was conducted and compared with the content of the draft plan. A comparison was made and the proportion of the results implementation in the draft plan was determined for the following structural elements: development baselines, objectives and guidelines; expert bases; stakeholders and public participation; sectoral interests; administrative competences; international dimension; and databases and analytical tools. A high degree of coherence was found, showing the obvious contribution of the projects at the methodological and structural levels.

**Keywords:** maritime spatial planning; Northern Adriatic; international projects; structural elements; first generation of maritime spatial plans

## 1. Introduction

In the countries of the European Union (EU), the implementation of maritime spatial planning is carried out in different ways and diverse administrative contexts. The processes are largely based on previous experiences in marine environment planning and conventional spatial planning practices. With the adoption of the MSP Directive (Directive 2014/89/EU of the European Parliament and of the Council of 23 July 2014 establishing a framework for maritime spatial planning) [1] and the ICZM Protocol (Protocol on Integrated Coastal Zone Management (ICZM) in the Mediterranean, 2008) [2], the Northern Adriatic countries have also begun formal procedures for the preparation of the first generation of maritime spatial plans (hereinafter referred to as "the plan").

The northern part of the Adriatic Sea is shared by Slovenia, Italy, and Croatia. These countries have similar historical starting points and development and environmental challenges [3,4]. Slovenia occupies the southern part of the Gulf of Trieste, where several administrative and political systems (Kingdom of Italy (1861–1946), Austro-Hungarian Empire (1867–1918), Kingdom of Serbs, Croats and Slovenes (1918–1929), German occupation in the World War II (1943–1945), Italian Republic (1946), Free Territory of Trieste under the auspices of the United Nations (1947–1954), Federal People's Republic of Yugoslavia

(1945–1963), Socialist Federal Republic of Yugoslavia (1963–1991), Republic of Slovenia (1991), and Republic of Croatia (1991)) have taken place over the past hundred years, leaving specific administrative, economic, and cultural impacts over the sea and land.

Despite many changes in state borders and pronounced multinationality, the region under consideration is a small, connected, and interdependent area in terms of spatial development. The regimes and uses of sea and coastal areas were largely formed in the second half of the previous century (post-war regulation of the state border between Yugoslavia and Italy), when a period of faster economic development began [5]. After the regulation of legal and political issues in 1954, the then Socialist Republic of Slovenia gained a sovereign access to the sea and established the foundations for modern maritime activities (beginning of the development of the Port of Koper) [6,7].

In the subsequent period, with the introduction of the doctrines of environmental protection and sustainable development, (Barcelona Convention [8], Brundtland Report [9], and European Spatial Development Perspective [10]) the area of the Northern Adriatic became a testing ground for research and implementation of sustainable principles in the field of activity planning at sea and on the coast [11]. Numerous activities began at the end of the previous century, ranging from transnational cooperation projects in the field of environmental protection and quality monitoring of the Adriatic Sea to the development of common approaches for MSP, involving all Adriatic countries due to the complexity of the marine environment.

The MSP Directive primarily represents a formal establishment of a common framework for the MSP in Europe [12,13], as marine spatial planning was considered long before within different international commitments (conventions, protocols, directives, etc.), national laws, and professional practices of EU countries in the field of marine environmental management and international preprojects, focusing on specific development, protection, and administrative challenges [4]. It is in the latter that several MSP parameters have gradually been formed, which are now being more precisely profiled and upgraded in the phase of the actual preparation of plans. Today, planners are faced with a vast legacy of professional references and practices that help to speed up the introduction of plans in countries or regions where such practices have not yet existed.

Given the short history of MSP, it can be assumed that European territorial and interterritorial cooperation programs (and individual focus projects within them) have significantly contributed to its development [14]. In 2010, the European Council adopted the EU Strategy 2020 for smart, sustainable, and inclusive growth, where coastal and maritime areas are identified as having the potential for sustainable growth. Within this perspective, MSP contributes to the strengthening of confidence and certainty for investors and thus to the implementation of the abovementioned strategy. The European Structural and Investment Funds, including the European Maritime and Fisheries Fund, provided funding to support MSP in the period 2014–2020. Contribution to the formation of MSP would not have been possible without the support from cross-sectoral cooperation projects, enabling the establishment of links between decision-makers, planners, spatial planning authorities, and the general public [14–16]. Such cross-border practices have upgraded the partial challenges of individual countries, enabled the balancing of interests in the wider area and ensured integrated management of the marine environment [17].

Development of the MSP process and structural elements of the plan has always been accompanied by several professional and scientific studies, addressing different international experiences and practices at legislative and methodological levels, mainly based on umbrella guidelines for the establishment of MSP (Convention on Biological Diversity—CBD COP-5 [18], Roadmap for Maritime Spatial Planning: Achieving Common Principles in the EU—EC COM [19], ICZM Protocol [2], and MSP Directive [1]) [20]. In this context the development of MSP can be monitored in three directions: (a) establishment of the MSP process, (b) plan preparation, and (c) MSP consolidation with the integrated coastal zone management (MSP-ICZM).

At the procedural level, special emphasis was placed on analytical approaches in setting up the MSP [12,21] and the ICZM process [22–26]. The aspects of sustainable development [26], international cooperation [27–29], and the ecosystem approaches [30–36] were highlighted in both cases (MSP and ICZM). The emphasis was also put on the correlation of land and sea planning systems [37–41] and the participation of stakeholders [42–47].

At the level of plan preparation, special emphasis was put on the correlation between the elements of conventional planning [37], methodology [48], classification and mobilization of stakeholders [49–52], sectoral interests [53], etc.

*Importance of International Projects in the Development of MSP Elements*

In the above-mentioned studies, the importance and contribution of international projects are repeatedly addressed in relation to a certain MSP segment. In the study of the EU legal framework and institutional support to maritime countries in setting up the MSP, Fries in Grémaud-Colombier [16] found that international projects play a key role in the process of establishing the international institutional interaction. The authors explain in detail the basic challenges and elements of the MSP, exposed in individual projects. Through the international projects, the European Commission has also financed and established permanent interstate cooperation mechanisms as an appropriate framework for effective balancing of interests and the exchange of good practices.

Ansong et al. [54] similarly explain the importance of international projects, summarizing their content and focus, and addressing their importance primarily in terms of developing partnerships, effective communication, and interinstitutional cooperation. The authors point to the differences between the projects realized before and after the implementation of the MSP Directive, since only the latter establishes a formal framework for defining the sectoral fields and other contents of a plan.

The role of international projects was also addressed in a specific survey conducted by de Grunt et al. [17], intended to determine the implementation of sustainable planning objectives. The survey was attended by practicing experts in the field of MSP development in EU countries. It was aimed at a hierarchical definition of sectoral fields, which, in the view of respondents, needed to be addressed extensively at the interstate level. Therefore, the international projects have proved to be an opportunity for the implementation of interstate communication in the direction of sectoral balancing (international workshops, mobilization of stakeholders, mobilization of land–sea interactions, etc.).

The cumulative significance and contribution of international projects is multilayered and reflected in the preparation phase of the first generation of the plan. According to the timeline for the implementation of the MSP Directive, all Northern Adriatic countries (and other EU countries) are currently in the active phase of plan preparation, drawing on experience, methodology, and data from all available sources. The first plan is definitely a unique innovation for individual countries.

Therefore, the relevance of the present research lies in the definition of the actual effect of international support projects. The research addresses the details of individual MSP elements, the process of their development and the possibility of their direct or indirect implementation. It also addresses the expediency or justification of the financial framework and the cooperation of the European Commission institutions.

The research results shall contribute to decision-making in further development and content of support projects both in the field of maritime and conventional spatial planning. It may also provide suggestions for the development of similar approaches to preliminary and/or cross-border coordinated planning in environments where such projects have not been or will not be implemented within the MSP process.

## 2. Materials and Methods

### 2.1. Problem Definition and Research Questions

Compared to neighboring Italy and Croatia, Slovenia, as a small maritime country, has less developed administrative instruments (sectoral legislation, professional practice,

institutions, staff, etc.); however, in recent decades, professional work has been carried out on its territory in the direction of MSP development, like in other EU countries. With the accession to the EU, Slovenia has adopted the EU legal order and adhered to all international commitments within its legislation (such as the Marine Strategy Framework Directive [55], the MSP Directive [1] and the ICZM Protocol [2]). The MSP implementation arrangements have also been carried out in the framework of other formal and informal activities, including in particular the participation in international support projects, which have contributed to the MSP development in various respects. In this context, the following questions are important for the present research:

- Which key international projects in the field of MSP development in the Northern Adriatic were implemented also in Slovenia?
- Which elements of the plan (Slovenia started preparing its first plan in September 2019. During submission of this article the draft of the plan (hereinafter referred to as "the draft plan") was in the phase of public consultation) were formed in individual projects?
- What is the contribution of such projects to the first generation of the plan?

*2.2. Methodology*

The research was conducted in four phases. In the first phase, materials that directly or indirectly deal with the maritime spatial planning and integrated coastal zone management (prepared after year 2000: final reports of international projects, minutes of workshops and working meetings, and reports of public presentations) were obtained. The materials were gathered from the archives of the Regional Development Centre of Koper, the Ministry of the Environment and Spatial Planning of the Republic of Slovenia, on the websites of individual projects, and from the producers of the draft plan.

In the second phase, at the beginning the materials were selected and a range of projects defined, which are analyzed in more detail below. The selection criterion was their contextual relevance in line with the provisions of the MSP Directive and the project terms of reference for the preparation of the draft plan. Next, seven key methodological and structural elements of the plan were identified on the basis of the MSP Directive provisions, previous research [41,56,57], and some recommendations [19], which provided the base for continuing the research.

In the third phase, descriptive and comparative research methods were used to perform a more detailed analysis of the content and consistency with the existing draft plan. This phase was conducted in two stages.

In the first stage, an analysis of the application areas of individual projects was performed, using the GIS tools to determine the extent of land and marine areas under consideration of each project. The surface areas (in hectares) and the rate of coincidence (%) with the draft plan were calculated. The GIS environment was established for the marine and land part of the Slovenian coast, where relevant vector spatial databases were collected, based on which it was possible to define the scope of all considered projects (land and sea borders of the Northern Adriatic countries, land–sea boundaries, boundaries between the internal waters and the territorial sea, different widths of coastal strips, protected areas of natural and cultural heritage, etc.).

In the second stage, seven elements of the plan were analyzed and compared, namely:

1. Content comparison of the analytical part of individual projects and the draft plan was performed and the rate of coincidence defined to assess compliance with international and national starting points, objectives, and guidelines.
2. Regarding the expert bases of individual projects, the extent (%) of their use in the preparation of the draft plan was determined.
3. Concerning the stakeholders and public participation, their pool and detailed structural classification was defined and the coincidence (%) with the draft plan determined for each project.

4. In assessing the identification and balancing of interests, a set of eleven fields, as defined in the MSP Directive, was used with the addition (in accordance with the draft plan) of urban development, covering other interests of local communities regarding the land–sea interactions. The presence and content of interests was determined and categorized into three levels. Moreover, the use of specific protocols and/or tools for the balancing of interests was identified.

5. Regarding the administrative competences, the focus was on spatial solutions of individual projects to find out how the sea and land areas were determined as to administrative competences and spatial planning authorities.

6. The international aspect of individual projects was addressed by recording the number of implemented events with international participation and categorized into three value groups. In addition, the participating countries and their institutions involved in the implementation of projects were identified.

7. The development of databases and the use of analytical tools were determined by analyzing the types of data used, the method of obtaining them, their accuracy and purpose, and processing in accordance with the project terms of reference. The extent of development of databases and analytical tools in individual projects was categorized into three value groups.

The fourth phase comprised the synthesis of the intermediate results and the drawing of conclusions.

When assessing the rate of coincidence, it was necessary to consider a certain deviation or tolerance (up to 5%), since it was not possible to establish adequate comparability for some elements (e.g., the assessment was very accurate in the stakeholder analysis, while specific generalizations were needed in the field of interest balancing). In a few cases, some elements could not even be compared. Some more detailed methodological steps are further substantiated in the presentation of results.

## 3. Results

### 3.1. Review of the Projects and Their Content Characteristics

In the process of material analysis, we identified seven international projects (Table 1), implemented also in Slovenia, and in which various elements of the draft plan appeared. The projects are: CAMP Slovenia [58], PLANCOAST [59], SHAPE [60], ADRIPLAN [61], ADRIATIC+ [62], SUPREME [63], and PORTODIMARE [64]. In line with the purpose and objectives, the projects included various horizontal activities (pilot projects, expert bases, thematic workshops, etc.).

### 3.2. Analysis and Comparison Elements

Maritime spatial planning is an institutional process, which results in a maritime spatial plan as a legal act (Spatial Planning Act of the Republic of Slovenia) [65]. Already in the process of developing this type of planning as a new professional practice, the content elements of the plan itself developed in parallel. In the content analysis of the material under consideration, a distinct intertwining of individual elements was encountered; therefore, it was difficult to precisely categorize the elements within the planning process domain or within the preparation and content domains of the plan. To this end, a structural model (Table 2) was developed based on the MSP Directive provisions, the experience of other authors and methodological recommendations [19]. The structural model classifies the basic elements of the planning process and the plan itself. It was found that three groups of elements could be distinguished, namely: procedural elements, methodological elements of the plan preparation, and content elements of the plan. Due to the focus of our research on determining the contribution of international projects to the preparation of the draft plan, the following seven elements were selected as the subject of our analysis: (1) starting points, objectives, and guidelines, (2) expert bases, (3) stakeholders and public participation, (4) identification and balancing of interests, (5) administrative competences,

(6) international aspects, and (7) databases and analytical tools. For these elements there was sufficient data in the materials to perform a comparative analysis.

**Table 1.** A list and the content outline of key international projects implemented also in Slovenia.

| Project Title (Implementation Period) Cooperation Program Participating Countries | Thematic Focus | Focus Activities and Results |
|---|---|---|
| 1. CAMP Slovenia (2004–2007) *Coastal Area Management Program* Mediterranean Action Plan (UNEP/MAP) In this case, not the individual countries participated, but the UNEP-MAP specialized Regional Activity Centres | Conception of an integrated approach to the management and sustainable development of the coastal zone. | Definition of the coastal zone as a specific spatial planning unit. Participative planning tool: system analysis and analysis of sustainable development perspectives. Production of vulnerability maps of the Slovenian Coast. Conceptual design of spatial arrangements of the coastal zone (expert basis). Comparison of management models for protected nature areas. Regional program for the protection of the environment and water resources. |
| 2. Plancoast (2006–2008) INTERREG IIIB CADSES Germany, Poland, Slovenia, Italy, Albania, Bosnia and Herzegovina, Bulgaria, Croatia, Romania, Serbia (including Montenegro) and Ukraine | Development of tools for integrated coastal and marine planning in the Baltic, Adriatic, and Black Sea regions. | Manual on integrated maritime spatial planning. Maritime spatial planning in selected areas (pilot project). Spatial plans for the coastal zone in accordance with the ICZM principles (pilot project). |
| 3. Shape (2013–2014) *Shaping an Holistic Approach to Protect the Adriatic Environment between coast and sea* AdriaticIPA 2007–2013 Italy, Slovenia, Croatia, Albania, Bosnia and Herzegovina, Montenegro | Strengthening the system of integrated coastal zone management. | Proposal for the definition of the coastal zone where construction is not possible in accordance with the ICZM Protocol (expert basis). Testing of the integration of maritime and land spatial planning in the Strunjan area, including a proposal for harmonized land use (pilot project). Geographic Information System (GIS) for the coastal zone. |
| 4. Adriplan (2014–2015) *Adriatic Ionian maritime spatial Planning* DG MARE, EASME Italy, Croatia, Slovenia, Greece | Support for the establishment of MSP in the Adriatic–Ionian Region, focusing on the cross-border aspects. | Formulation of proposals and recommendations for cross-border maritime spatial planning in the Adriatic–Ionian Region. The project is based on integrated assessment of the situation (environmental, legal and administrative bases, economy and society), considering the needs and potentials of the sectors. |

**Table 1.** *Cont.*

| Project Title (Implementation Period) Cooperation Program Participating Countries | Thematic Focus | Focus Activities and Results |
|---|---|---|
| 5. Adriatic+ (2016) AdriaticIPA 2007–2013 Italy, Croatia, Montenegro, Slovenia | Exchange of experience in the management of marine and coastal areas in the Adriatic region. | Exchange of experiences in the management of marine and coastal areas in the Adriatic region (thematic workshops). Capitalization of the results of three standard projects (NETCET (https://www.netcet.eu, accessed on 1 March 2021) SHAPE (https://www.msp-platform.eu/projects/shaping-holistic-approach-protect-adriatic-environment-between-coast-and-sea, accessed on 1 March 2021) and HAZARD (https://www.msp-platform.eu/projects/hazadr-strengthening-common-reaction-capacity-fight-sea-pollution-oil-toxic-and, accessed on 2 March 2021) and two strategic projects (BALMAS (www.balmas.eu, accessed on 2 March 2021), DEFISHGEAR (http://adriplan.eu/, accessed on 2 March 2021). Feasibility study to set up a decision support system for biodiversity protection measures. |
| 6. Supreme (2018–2019) *Supporting maritime spatial Planning in the eastern Mediterranean* DG MARE, EASME Italy, Croatia, Greece, Slovenia | Simulation of the procedure and content of MSP—preparation of materials and the first plan. | Cross-section of the situation in the field of available data and interests. Preparation of materials for the plan. Simulation of the MSP process and content (pilot project). Compilation of sectoral and international starting points and objectives. Pool of stakeholders, balancing of interests, thematic workshops. |
| 7. Portodimare (2018–2020) *Geoportal of Tools and Data for Sustainable Management of Coastal and Marine Environment* Interreg ADRION Italy, Croatia, Greece, Slovenia, Montenegro, Bosnia and Herzegovina | Development and testing of tools and methods to support the implementation of MSP, establishment of a common geoinformation infrastructure. | Upgrading of the SHAPE (http://www.shape-ipaproject.eu/, accessed on 2 March 2021) and ADRIPLAN (http://adriplan.eu/, accessed on 2 March 2021) projects results. Creation of a common information platform (Geoportal). Development of analytical tools for the plan. Testing of selected analytical tools. |

**Table 2.** Key elements for the development of the maritime spatial planning process, methodology of the plan preparation, and contents of the plan.

| Maritime Spatial Planning Development | | |
|---|---|---|
| **A**<br>**Maritime Spatial Planning Establishment of the Process** | **B**<br>**Maritime Spatial Plan**<br>**Spatial Act** | |
| | ↓<br>**Process of the Maritime Spatial Plan Preparation** | ↓<br>**Content of the Maritime Spatial Plan** |
| **Process Elements** | **Methodological Elements** | **Content Elements** |
| Promotion of international commitments (conventions, protocols).<br>Rationale for the need to develop maritime spatial planning and integrated management of the coastal zone.<br>Implementation of sustainable planning principles.<br>Promoting the need for participatory planning.<br>Implementation in national legislation. | Definition of starting points.<br>Identification of stakeholders and methods of participation.<br>Development of a database and Geoportal.<br>Expert bases.<br>Setting of strategic and planning objectives.<br>Balancing of interests (range and methodology).<br>International aspects (coordination).<br>Administrative competences.<br>Monitoring and evaluation. | Formal framework (mandatory content).<br>Rationale of MSP (starting points and objectives).<br>Sectors (in line with the MSP Directive).<br>Regimes and uses of maritime space (identification).<br>Spatial and time coordination of the regimes and uses of maritime space.<br>Methods of implementation (provisions, guidelines, protocols).<br>Mapping (database). |

### 3.3. Analysis of Project Application Areas

In the analysis of the spatial extent of application areas, it was found (depending on the purpose) that individual projects had treated water and land areas with different accuracy.

The analysis covered (a) a detailed treatment (analysis of protection regimes or the arrangements for the implementation of a specific activity, identification of interests and conflicts, determination of parcel boundaries, etc.) and (b) a general treatment, considering the overall importance of the areas under consideration (interactions and links between sectors, broader economic importance of a particular activity at sea and on land, etc.). Based on a detailed drawing of application areas (areas covered by expert bases, conceptual spatial solutions, pilot projects, etc.), some overlapping was found, which shows a multiple treatment of the same area. In the projects, the Slovenian sea was divided into the territorial sea and the internal waters of the Republic of Slovenia, while the Strunjan Bay was considered separately. The coastal zone was divided into three segments of 100 m, 200 m and 500 m strips, and additionally to the wider hinterland of the Strunjan Bay and the protected areas extending beyond the coastal strip. The projects implemented before the international arbitration award (the sea border between the Republic of Slovenia and the Republic of Croatia was determined by the international arbitration award of 29 June 2017) covered a larger territory at sea; however, this research is limited to the reach after arbitration, which is also considered in the current draft plan (Table 3). The following findings were made:

- The internal waters were discussed in detail in each project (with the exception of the ADRIPLAN project).
- The territorial sea was discussed in detail in two projects and was the subject of general consideration in five.
- The coastal strip of 100 m from the shore was treated in projects four times and the strip of 200 m three times. The coastal strip in additional widths of 300 m and 400 m was considered in general three times.

**Table 3.** Application areas of individual projects, surface of the project application areas, and the share of total area in relation to the maritime spatial plan area.

| Project | Sea (Outwards from the Shoreline) | | Coast (Inwards from the Shoreline) | | Σ Sea | Σ Coast | Σ |
|---|---|---|---|---|---|---|---|
| | Detailed Research of Regimes and Uses | General Research of influence Areas | Detailed Research of Regimes and Uses | General Research of Influence Areas * | | | |
| | (Area in ha) | | | | Percentage Compared to MSP | | |
| CAMP Slovenia | Internal waters (4973.17) | Territorial sea (16,530.02) | 200 m strip (797.19) | 300 m strip; protected areas *** (1988.30) | 100.00% | 143.99% | 104.18% |
| PLANCOAST | Internal waters (4973.17) | Territorial sea (16,530.02) | 100 m strip (411.95) | / (0.00) | 100.00% | 21.30% | 93.48% |
| SHAPE | 200 m strip; the entire Bay of Strunjan (864.18) | Territorial sea (20,642.54) | 200 m strip; saltpans area; hinterland area of the Strunjan settlement (1522.86) | 300 m strip; protected areas *** (1406.68) | 100.00% | 151.44% | 104.82% |
| ADRIPLAN | / (21,378.62) | Entire sea ** (0.00) | / (0.00) | / (0.00) | 100.00% | 0.00% | 91.70% |
| ADRIATIC+ | Internal waters (4973.17) | Territorial sea (16,530.02) | / (1412.29) | / (0.00) | 100.00% | 0.00% | 91.70% |
| SUPREME | Entire sea ** (21,378.62) | / (0.00) | 200 m ICZM strip (941.4) | 400 m strip; protected areas *** (1831.70) | 100.00% | 143.35% | 103.60% |
| PORTODIMARE | Entire sea ** (21,378.62) | / (0.00) | / (0.00) | / (0.00) | 100.00% | 0.00% | 91.70% |
| Draft plan | Entire sea ** (21,378.62) | / (0.00) | ICZM strip protected areas *** (1934.46) | / (0.00) | 100.00% | 100.00% | 100.00% |

* General research: influence area (detailed study of the strip width and further inwards). ** Internal waters and territorial sea. *** Protected areas of natural and cultural heritage (up to 500 m from the coastline).

Compared to the scope of the draft plan, some land-based areas were addressed to a greater extent (CAMP Slovenia (104.18%), SHAPE (151.44%), and SUPREME (143.35%)), with the CAMP Slovenia project comprising a broader regional framework in which the various horizontal projects covered the hinterland areas (not considered in the above calculation).

Figure 1 and Table 3 show that the application areas of all projects are quite similar. They all deal with the entire national sea, the only difference being in the extent of treatment in relation to the focus of each project. However, larger differences occur on land. In particular, the sizes of areas differ, which is expected to have a greater impact on the interaction between land and sea in the coastal zone. The interest areas of three projects (ADRIPLAN, ADRIATIC+, and PORTODIMARE) do not cover the mainland. It can thus be concluded that both the marine and land areas were studied several times from different aspects even before the development of the maritime spatial plan. On the basis of the above, it can be concluded that in Slovenia, compared with the scope of the draft plan, all projects have been implemented in full, which cannot be said for other countries. This is due to the relatively small water and land areas, and similar geographical, economic, and social characteristics of the functionally closely linked areas. From this point of view, the decision to analyze each case in an entirety is logical and consistent with the objectives of the individual projects.

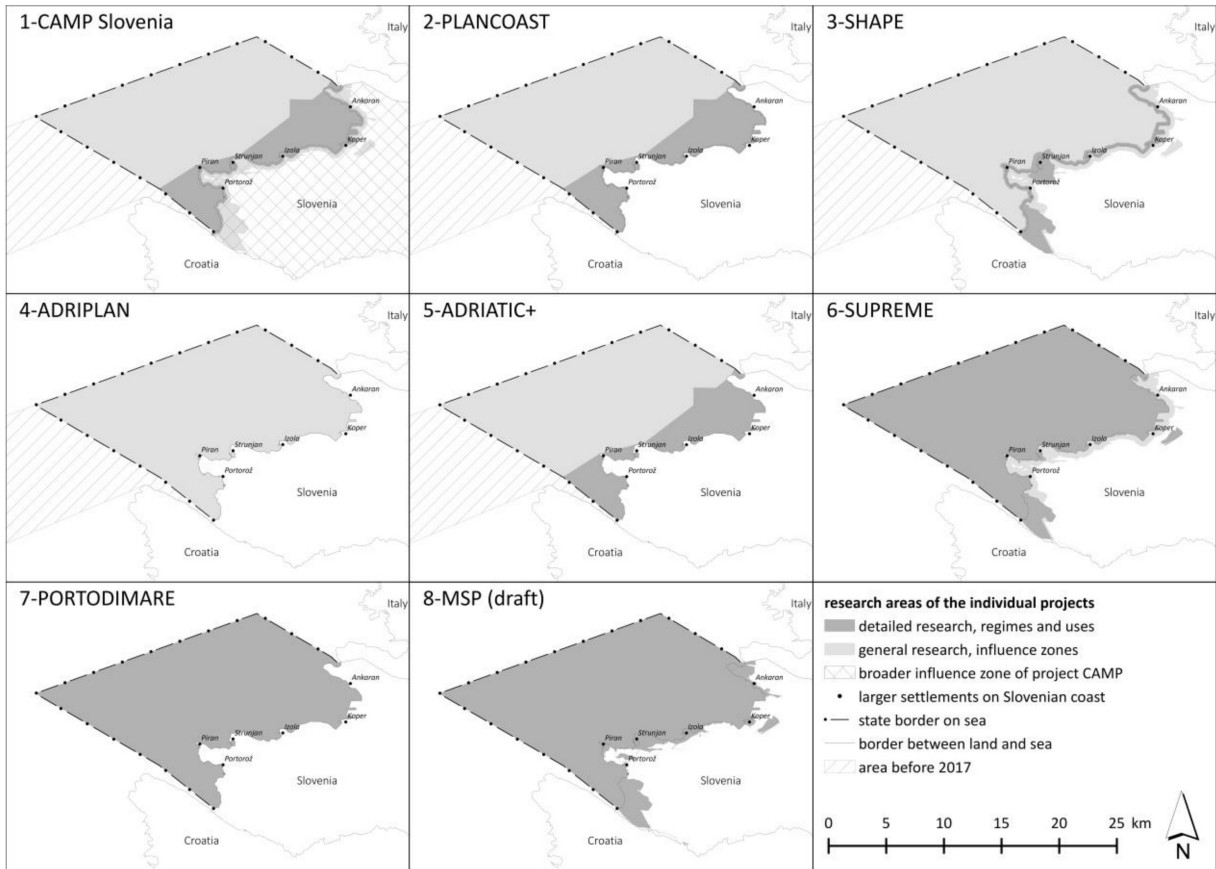

**Figure 1.** Project application areas; synthetic presentation of particular studies.

*3.4. Analysis of Methodological and Structural Elements, and the Assessment of Consistency with the Draft Plan*

3.4.1. Starting Points, Objectives, and Guidelines

In the analytical work, all the considered projects summarize the contents of various international and national documents and acts. Based on the analysis of the material, five recurring groups of documents were identified, which starting points, objectives, and guidelines were found within the contents of the analyzed projects. These are: (1) international documents in the field of protection or development of the marine environment (European and international conventions, strategies, and directives, e.g., Barcelona Convention [8], etc.), (2) national spatial development and protection documents (e.g., Spatial Development Strategy of Slovenia [66], Marine Environment Management Plan [67], etc.), (3) local spatial documents (municipal strategic and implementing spatial acts, etc.), (4) sectoral development strategies at the national and municipal levels (e.g., Fisheries Management Plan, Landscape Park Management Plan, Regional Development Program [63], etc.), and (5) local initiatives (materials obtained in the framework of the preparation of municipal spatial plans, fishermen, mariculture and tourism initiatives, etc.).

The observation of starting points, objectives, and guidelines in the above-mentioned documents varies considerably among the projects. In some cases, it is only a matter for recording or quoting the documents, while in others it involves strict consideration when designing concrete spatial or strategic solutions (e.g., sectoral objectives, objectives in reconciling interests at sea and on land, compliance with nature protection guidelines, etc.).

Comparatively with the draft plan (i.e., which contents and to what extent they are considered), it was found (Table 4) that in regards to the provisions of international documents, the compliance was very high or almost 100% in four out of seven projects. This is because the projects were financially assisted from international programs, indented to

support the implementation of international strategic documents and legal order. Moreover, the authors of these projects were experts with relevant experience in the field of sustainable development and spatial planning, well acquainted with this kind of materials and, consequently, they consistently implemented them in their professional work. There is a slightly weaker coincidence noted in early projects (CAMP Slovenia and PLAN-COAST), as some of the commitments presently in force did not exist at the time of their implementation (e.g., Marine Strategy Framework Directive, 2008 [55] and MSP Directive, 2014 [1]). The same applies to the compliance with documents at national and local levels, except for the ADRIATIC+ project, which did not focus on specific local development and protection challenges, and the PORTODIMARE project where the national content was irrelevant, since the project was developing the Geoportal and methodological tools for the Adriatic–Ionian Region.

**Table 4.** Comparison of the draft plan and individual projects relating to their consistency with the starting points, objectives, and guidelines of relevant international and national documents and commitments.

| Starting Points, Objectives and Guidelines (Identification and Consistency) | Camp Slovenia | Plancoast | Shape | Adriplan | Adratic+ | Supreme | Portodimare |
|---|---|---|---|---|---|---|---|
| International documents for the sea and coast | 55% | 80% | 95% | 100% | 100% | 100% | 100% |
| National strategic spatial planning documents | 50% | 80% | 100% | 100% | n.r. | 100% | n.r. |
| Local spatial planning documents | 90% | 90% | 100% | 50% | n.r. | 100% | n.r. |
| Sectoral development needs (11 areas) | 30% | 30% | 25% | 100% | n.r. | 85% | 30% |
| Local initiatives: identification and compliance (public and private investors) | 90% | 30% | 25% | 50% | n.r. | 95% | n.r. |

Draft plan = 100%, n.r. = not relevant.

It applies to all projects that different starting points, objectives, and guidelines were noted, as also considered to some extent, within their preparation. Within this analysis, only their presence was recorded, while their consistency and the effect of their consideration were, however, not established. Nevertheless, it can be concluded that multiple or long examination of individual starting points, objectives, and guidelines strengthens their promotion among the decision-makers and in the general public.

3.4.2. Expert Bases

Concerning the expert bases preparation for the needs of the implementation of individual projects, it was found that certain new professional support materials were produced in all projects. In view of project content orientation and objectives, the materials were divided into two groups: (a) sectoral expert bases, separately for sea and land, and (b) integrated spatial solutions or orientations, separately or jointly for land and sea. The materials include pilot projects, conceptual solutions, and detailed spatial solutions and planning guidelines, except for the PORTODIMARE project, which contributed a common data platform (Geoportal) and information and decision support tools focused and tested on entire national coastal and maritime areas.

In relation to the draft plan, detailed sectoral expert bases and integrated spatial solutions, for the land and partly also for the sea, were developed already within the CAMP Slovenia project. The same applies to the SHAPE project. Within the ADRIPLAN project, a detailed analysis of the regimes and uses in the coastal zone was elaborated. The SUPREME project was specific, as it contributed an integrated professional basis for the draft plan, both in procedural (methodological) and content terms. All expert bases were completely cofinanced, managed, and coordinated within these projects. Thus, their results also represent a significant financial, time, and methodological contribution to the process of content preparation for the first draft plan.

A comparative analysis of the draft plan content highlighted the extent to which the draft plan summarizes the expert bases made within individual projects (Table 5).

It was established that the expert bases were completely incorporated in relation to the 100 m ICZM coastal zone (SHAPE expert bases) and up to 90% regarding the definitions of sectoral uses at sea and land (SUPREME expert bases). Moreover, there is a wide range of other materials, which were considered to a lesser extent or only partially, but certainly had a significant impact on the comprehensive treatment of sectoral fields and spatial solutions, as later defined in the draft plan (interests, constraints, spatial characteristics, etc.).

### 3.4.3. Stakeholders and Public Participation

Identification and comparison of the participating stakeholders and the participating public was carried out by analyzing those invited to (a) working meetings (b) coordination workshops, and (c) interim and final public presentations of the projects, distinguishing two basic groups, namely: the explicitly invited stakeholders (involved in the project development process) and those who participated on their own initiative (mostly in public presentations). Stakeholders were recorded in all projects. According to their institutional affiliation, they were divided into six groups: (1) sectoral representatives at the national level, (2) representatives of local communities (municipalities), (3) representatives of spatial planning bodies, (4) representatives of the economy, (5) external professional public, and (6) general public. For each group, their interest in participation (administrative competence, professional duty, development or protection interests, public interest, and personal interest) and the ways of participation (providing guidelines or opinions and attending the meetings, workshops, and public presentations) were identified (see example in Table 6).

It was found that since the time of the CAMP Slovenia project, the pool of invited and participating stakeholders was, to a certain extent, recurring and supplementing itself (Table 7). Although the number of participants in project varied, their interest, and institutional structure was maintained. This was mainly due to the relatively limited national set of competent institutions and their representatives, and a limited number of motivated individuals from other interest groups (initiatives, associations, external experts, etc.). It was also found that representatives of all coastal municipalities participated in all projects (100%), while the representation of ministries, as compared to the participation in the draft plan, was the highest in the ADRIPLAN (88%) and SUPREME (88%) projects. In the group of other stakeholders, the SHAPE (80%) and SUPREME (80%) projects come closest to the draft plan and the CAMP Slovenia (147%) project even exceeds it, mainly because it covered a wider inland territorial area and other issues. Namely, a number of representatives of the general and professional public, who were not necessarily linked to the sea and the coastal zone, also took part in the CAMP Slovenia Project (22 stakeholders).

**Table 5.** Comparison of the draft plan and individual projects: various expert bases for land and sea.

| Expert Bases | Camp Slovenia | Plancoast | Shape | Adriplan | Adratic+ | Supreme | Portodimare |
|---|---|---|---|---|---|---|---|
| Sectoral—land | 70% | 0% | 100% | 0% | 25% | 90% | n.r. |
| Sectoral—sea | 30% | 0% | 25% | 20% | 25% | 100% | n.r. |
| Integrated spatial guidelines and solutions—land | 70% | 60% | 100% | 0% | n.r. | 95% | n.r. |
| Integrated spatial guidelines and solutions—sea | 30% | 25% | 25% | 80% | n.r. | 95% | n.r. |

Draft plan = 100%, n.r. = not relevant.

**Table 6.** Detailed analysis of the participating stakeholders in the SUPREME project, prepared based on the minutes of the workshops, working, and coordination meetings and the final report.

| Stakeholders (Supreme) | Participation Criteria | | | | | Way of Participation | | | | Number of Invitees (Supreme) | | Number of Invitees (Draft Plan) | |
|---|---|---|---|---|---|---|---|---|---|---|---|---|---|
| Example: Supreme Project Draft Plan = 100% ●Record of Participants | Administrative Competence | Professional Duty | Development and Protection Interests | Public Interest | Personal Interest | Providing Guidelines | Giving Opinions | Participation in Meetings/Workshops | Participation in Public Presentations | Number of Institutions | Number of Individuals | Number of Institutions | Number of Individuals |
| 1. Representatives of sectors at the national level (ministries) | ● | | ● | | | ● | ● | ● | ● | 7 | 9 | 8 | 39 |
| 2. Representatives of local communities (municipalities) | ● | | ● | | | ● | ● | ● | ● | 4 | 9 | 4 | 8 |
| 3. Spatial planning bodies | ● | | ● | | | ● | ● | ● | ● | 2 | 2 | 7 | 15 |
| 4. Representatives of the economy | | ● | | | | ● | ● | ● | ● | 6 | 6 | 4 | 6 |
| 5. External professional public | ● | ● | | ● | | | ● | ● | ● | 2 | 3 | 2 | 2 |
| 6. General public | | | | ● | ● | | ● | | ● | 2 | 3 | 2 | 2 |
| Total | | | | | | | | | | 23 85% | 32 44% | 27 100% | 72 100% |
| Total number of institutions | 1 = 88%, 2 = 100%, 3 + 4 + 5 + 6 = 80% | | | | | | | | | | | | |

**Table 7.** Comparison of the draft plan and individual projects: participating stakeholders (sectors, local communities, and other stakeholders); the basis for the calculation are the invitees and participants in the draft plan (100%).

| Stakeholders and Public Participation | Camp Slovenia | Plancoast | Shape | Adriplan | Adratic+ | Supreme | Portodimare |
|---|---|---|---|---|---|---|---|
| Sectors (ministries) at the national level (draft plan = 8) | 50% | 13% | 22% | 88% | 13% | 88% | 13% |
| Local communities (municipalities (draft plan = 4) | 100% | 100% | 100% | 100% | 100% | 100% | 100% |
| Other stakeholders (spatial planning bodies, external professional public, general public) (draft plan = 15) | 147% | 20% | 80% | 53% | 13% | 80% | 40% |

Draft plan = 100%, Note: The shares take account of the number of institutions or individuals of different backgrounds.

### 3.4.4. Identification and Balancing of Interests

It was established in the research that balancing of development and protection interests was carried out in the framework of: (a) preparation of expert bases and conceptual projects (in discussion with relevant stakeholders), (b) focus meetings with the sectoral representatives, and (c) workshops with extended participation. It is evident from the analyzed material that balancing of interests was carried out to a different extent and with different purposes, but the results are difficult to accurately assess based on the available data. The analysis therefore focused on identifying a range of individual sectors (sectoral interests covering the sea and land), which were the subject of individual projects (Table 8). Based on the interim results, there were three categories of treatment identified:

(a) Explicit treatment comprising a detailed definition of sectoral characteristics and interests (Figure 2a,b).

(b) Indirect treatment consisting of partial identification and/or indirect consideration of sectoral interests.

(c) Sectors and their interests are not explicitly or indirectly defined.

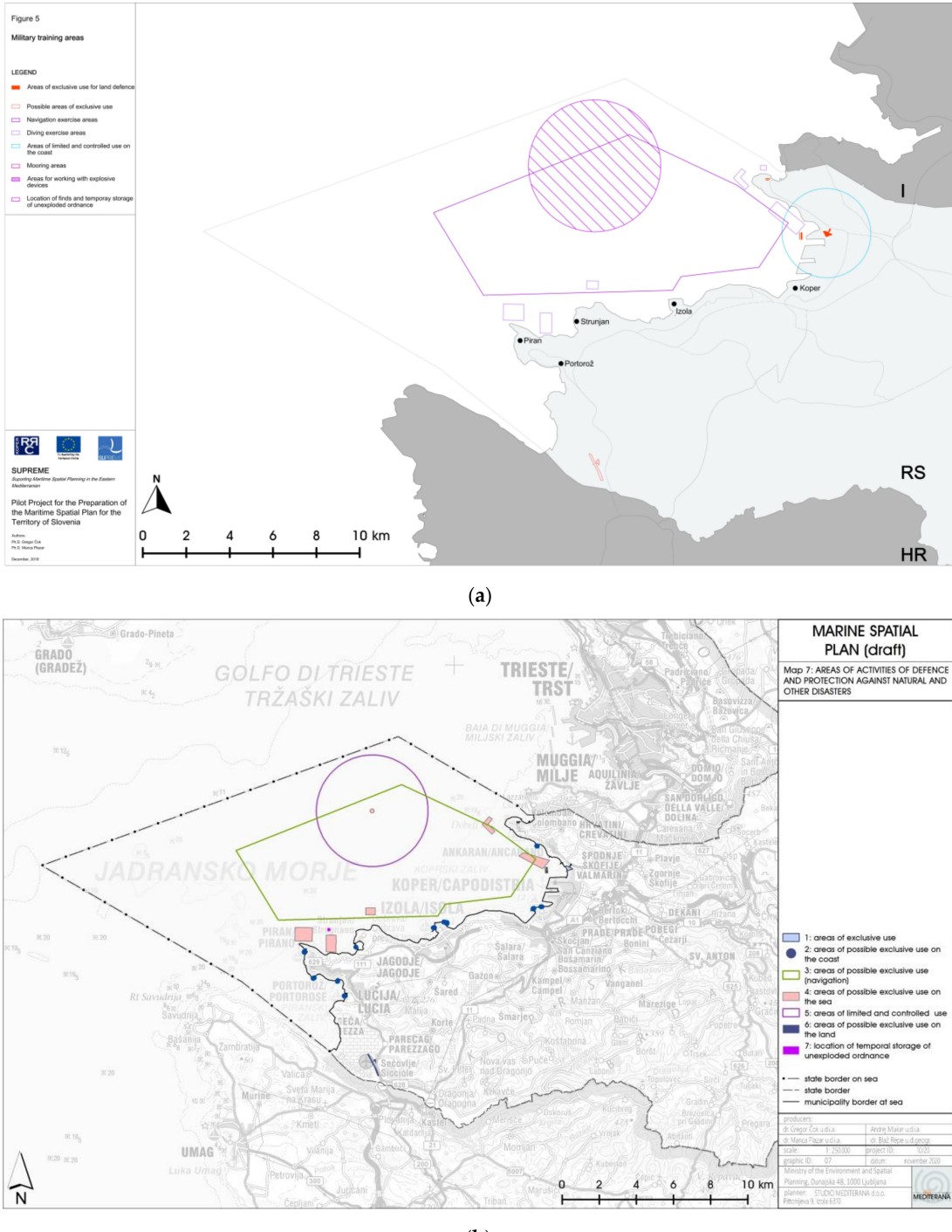

(**a**)

(**b**)

**Figure 2.** Identification of military training areas: SUPREME [63] (**a**) and draft plan [68] (**b**). The figures illustrate high coherence between the SUPREME project and the draft plan.

**Table 8.** Comparison of the draft plan and individual projects: identification and treatment of sectoral interests classified in 11 + 1 fields, as defined by the MSP Directive.

| Identification and Balancing of Interests | Camp Slovenia | Plancoast | Shape | Adriplan | Adratic+ | Supreme | Portodimare |
|---|---|---|---|---|---|---|---|
| Mariculture (sites and infrastructure) | o | + | + | + | + | + | + |
| Fisheries (fishing areas and fishing infrastructure | o | + | o | + | o | + | + |
| Exploration and exploitation of oil, gas and other energy sources | - | - | - | - | - | + | + |
| Maritime transport (routes and traffic flows) | + | + | + | + | + | + | + |
| Defense (military training areas) | - | - | - | - | - | + | + |
| Nature protection (protected areas) | + | + | + | + | + | + | + |
| Exploitation of raw materials | + | + | + | - | + | + | + |
| Scientific research | o | - | - | - | o | + | + |
| Submarine cables and pipelines | - | - | - | + | - | + | + |
| Tourism (tourism infrastructure) | + | + | + | + | + | + | + |
| Cultural heritage (underwater and coastal heritage) | + | + | + | + | + | + | + |
| Urban development (local land–sea interests) | + | + | + | + | + | + | - |
| Identification and application of protocols and tools for the balancing of sea/land interests | *** | *** | *** | ** | ** | *** | *** |

(+) Explicit treatment, (o) indirect treatment, (-) not defined; (***) full application, (**) partial application.

The presence or application of specific protocols and/or tools for the balancing of interests was identified in the same way.

It can be concluded that conventional sectors (mariculture, fisheries, maritime transport, tourism, and protection of natural and cultural heritage) were dealt with mostly all projects, as also the interests of coastal local communities. The interests of seabed research, scientific research and defense were treated less often, which is a logical consequence of the size of the Slovenian sea and legal restrictions. The use of specific tools for the balancing of interests was also identified in all projects.

3.4.5. Administrative Responsibilities for the Sea and Land

In Slovenia, according to the legislation, the state has full authority and responsibility over the sea and the municipalities have jurisdiction over the land, but in the 25 m wide strip of land, defined as the coastal land of the sea, the state imposes many land-use restrictions. Such a division of administrative jurisdiction makes integrated planning and management challenging. Administrative responsibilities were taken into consideration within the analysis of the results and implementation methodology of individual projects. However, the projects repeatedly pointed out this issue, especially when drawing up certain spatial design solutions. This structural element of the MSP is evident in the phase of defining spatial planning units, based on a specific spatial act or in the call of competent spatial planning authorities to provide project guidelines (see Section 3.4.3). Therefore, the method of defining spatial planning units and identification of competent institutions were determined along the following lines (Table 9):

(a) Definition of water area as a specific spatial planning unit (e.g., parcels of territorial sea or internal waters).

(b) Definition of the coastal zone as a specific spatial planning unit, covering the area from the shoreline inwards.

(c) Identification of spatial planning authorities and their administrative competences.

**Table 9.** Comparison of the draft plan and individual projects: identification and examination of administrative spatial entities and administrative authorities.

| Administrative Authorities for the Sea and Land | Camp Slovenia | Plancoast | Shape | Adriplan | Adratic+ | Supreme | Portodimare |
|---|---|---|---|---|---|---|---|
| Definition of the water area as a specific spatial planning unit | + | + | o | + | o | + | o |
| Definition of the coastal zone as a specific spatial planning unit | + | + | + | - | + | + | - |
| Definition of land–sea administrative responsibilities (or proposal of revision of the existing) | + | o | + | - | + | + | - |

(+) Explicitly defined, (o) partially defined, (-) not defined.

Most projects followed the definition of spatial planning units at sea and on land (except for ADRIPLAN and PORTODIMARE) in accordance with the applicable spatial planning acts and the provisions of spatial planning authorities. In the SHAPE project, the ICZM coastal zone (a special expert basis [69] was prepared dealing with the issue of adaptation of Article 8 of the ICZM Protocol to Slovenian conditions. This article imposes on the States Signatories to the Protocol to establish a base 100 m coastal zone on land where construction is in principle not possible. The proposal extended this zone with areas that are not buildable under other regulations. Moreover, a coordination structure was proposed for the extended zone, which reaches also into the sea. Part of the expert basis also deals with the issue of more effective coordination between state departments (jurisdiction over the sea) and municipalities responsible for the land part of the coastal zone) was precisely defined as an area of special administrative treatment (in coordination with spatial planning authorities). At the same time, most projects also highlighted the need for a clear definition of responsibilities in the field of land–sea integrated planning. Within this framework, several proposals was identified (among the project results) for the revision of existing responsibilities to be formalized by establishing a maritime spatial plan (e.g., granting competences in the management of a 100 m coastal zone to the coastal municipalities).

3.4.6. International Aspect

The projects were implemented under territorial cooperation programs (and other EU Commission programs, and the UNEP-MAP, thus providing an opportunity to establish cross-border partnerships at different administrative levels (Adriatic–Ionian Region, Mediterranean Region and global UNEP-MAP). In addition to the focal goals, it is evident from the content of the of the considered projects that their essence is also the maintenance of contacts with science and modern practices in the world, implementation of up-to-date approaches and methodologies, developed within international organizations (e.g., UNEP-MAP) or joint development of new approaches, methodologies, tools, and databases.

In addition, the international aspect was considered within the implementation of international commitments in the form of directives, protocols, multinational agreements, etc. (see Section 3.1). Detailed contents and the effect of considered international aspects (except for international starting points) are difficult to assess analytically and accurately based on available data. Therefore, in the analysis of international aspects, the following was carried out based on collected materials: (a) identification of the number of events with international participation (meetings, workshops, presentations, etc.), which represented an opportunity to coordinate project contents, intermediate results, and the promotion of results (Table 10), and (b) identification of the participating partners and their institutional classification (Tables 11 and 12).

**Table 10.** Number of events with international participation in individual projects.

| International Aspect | Camp Slovenia | Plancoast | Shape | Adriplan | Adratic+ | Supreme | Portodimare |
|---|---|---|---|---|---|---|---|
| Thematic meetings with international participation | *** | ** | ** | ** | * | ** | ** |
| Focus workshops with international participation (coordination) | *** | * | ** | * | * | * | * |
| Public presentations of project results with international participation | *** | ** | * | * | * | * | * |

Number of events: 1–3 (*), 3–5 (**), 5+ (***).

**Table 11.** Range of institutions of the participating countries in individual projects—synthesis presentation of the total number.

| Project partner/institution | Camp Slovenia | Plancoast | Shape | Adriplan | Adriatic+ | Supreme | Portodimare |
|---|---|---|---|---|---|---|---|
| 1. Local level (municipality) | 9 | 1 | 0 | 1 | 1 | 0 | 0 |
| 2. Regional level institution | 1 | 6 | 8 | 9 | 3 | 1 | 5 |
| 3. National level (ministry) | 6 | 2 | 1 | 0 | 0 | 3 | 0 |
| 4. Sectoral agency | 4 | 1 | 2 | 0 | 0 | 0 | 2 |
| 5. Non-governmental organization | 1 | 1 | 0 | 0 | 1 | 0 | 0 |
| 6. Research institution | 3 | 3 | 0 | 6 | 3 | 6 | 3 |
| 7. Enterprise (professional) | 0 | 1 | 0 | 1 | 0 | 0 | 0 |
| 8. International organization (UNEP-MAP) | 6 | 1 | 1 | 0 | 0 | 2 | 1 |
| TOTAL number of partners | 30 | 16 | 12 | 17 | 8 | 12 | 11 |

**Table 12.** List of participating institutions in the case of the PORTODIMARE project.

| Portodimare Project Partners | Institution Name and Country |
|---|---|
| Regional level institution | Emilia-Romagna Region, Directorate General for Territory and Environment Protection (IT) <br> Abruzzo Region, Service for Maritime Works and Marine Water (IT) <br> Apulia Region Department of Civil Protection (IT) <br> Veneto Region, Environment Directorate—Integrated Water Service and Water Protection Unit (IT) <br> Regional Development Centre Koper (SLO) |
| Research institution | CORILA—Consortium for Managing Research Activities in the Venice Lagoon (IT) <br> Hellenic Centre for Marine Research, Institute of Marine Biological Resources and Inland Waters (GRE) <br> Institute for Physical Planning, Region of Istria (CRO) |
| Sectoral Agency | Public Enterprise for Coastal Zone Management of Montenegro (MN) <br> Centre for Economic, Technological and Environmental Development Sarajevo (BIH) |
| International organization | Priority Actions Program Regional Activity Centre (CRO) |

IT—Italy, GRE—Greece, CRO—Croatia, SLO- Slovenia, MN—Montenegro, BiH—Bosnia and Herzegovina.

Regarding the implemented events, the CAMP Slovenia project stands out, as within it several activities was carried out with the participation of the UNEP-MAP experts and their Regional Activity Centres. After the project completion, many presentations were held at the occasions of events and elsewhere within the UNEP-MAP structure (promotion). In other projects there were many thematic meetings. This demonstrates the intensive transnational cooperation from the very beginning of the establishment of the ICZM and MSP concepts in Slovenia and the Northern Adriatic.

As for the participating partners, there were eight different groups of institutions identified (Tables 11 and 12), which include different administrative levels (local, regional, and national) and fields of work (research, professional, and business interests). The range of participating partners represents a wide potential for establishing international institutional partnerships, based on which Slovenia can effectively integrate in the management of strategic challenges in the Adriatic–Ionian Region (EU Strategy for the Adriatic–Ionian Region—EUSAIR [70]). This was also confirmed by participation of the UNEM-MAP (PAP RAC) representatives in the preparation of the draft plan.

3.4.7. Development of Databases and Analytical Tools

Concerning the information support development to the MSP process, there were three levels identified, namely: data collection, establishment of a database (Geoportal), and the development of analytical tools (Table 13). The predominant types of data concern (a) land use and sea use, (b) protection regimes and regimes for the implementation of selected activities on land and sea, and (c) other spatial and statistical data (tourism, demography, socioeconomic data, ecology, energy, infrastructure, etc.). Despite the diversity of the projects and the accuracy of the individual data, a common attribute was present at all projects' final stages: a synthesis database was created for the needs of project activities. In most cases, data infrastructure (Atlas, Geoportal) supporting the decision-making was one of the results of the projects (especially in ADRIPLAN, SHAPE, SUPREME, and PORTODIMARE projects), which was put into general use after the completion of projects.

**Table 13.** Formed and used databases and analytical tools.

| Databases and Analytical Tools | Camp Slovenia | Plancoast | Shape | Adriplan | Adratic+ | Supreme | Portodimare |
|---|---|---|---|---|---|---|---|
| Database formed for the needs of project implementation | + | + | + | + | o | + | + |
| Geoportal formed (as a project module), publicly available database | o | + | + | + | o | + | + |
| Formed analytical tools (as a project module), publicly available for the purpose of carrying out analyses and spatial solutions | + | - | - | + | + | + | + |

(+) fully formed, (o) partially formed (-) not formed.

Analytical tools were developed in five projects (except for PLANCOAST and SHAPE).

It is evident that various data and analytical tools were upgraded quantitatively and qualitatively from project to project. From this point of view, it is not possible to determine the exact share of their use in the draft plan, as it was based on the data provided by the contracting authority (Ministry of the Environment and Spatial Planning) and represents a synthesis of all previous activities in the field of the development of databases and analytical tools.

*3.5. Synthesis of Findings*

A detailed analysis of the identified projects shows that at the beginning of the preparation of the draft plan (in Slovenia in September 2019) all key elements were already addressed to a certain extent. In the implementation of the individual projects or their expert bases, the following elements appeared several times:

1. Identification of the starting points of international conventions, protocols, directives, and recommendations and national platforms for spatial development and protection, and the objectives set out in the overarching strategic national documents, etc.
2. Creation of thematic expert bases and conceptual projects, their application and identification of the needs for additional expert bases for the preparation of the draft plan.

3. Identification and involvement of stakeholders and decision-makers, active participation, and public presentations.
4. Identification of interests and conflicts and demonstration of their mitigation (focal point in the CAMP Slovenia, SHAPE, ADRIPLAN, SUREME, and PORTODIMARE projects).
5. Addressing the administrative challenges in the field of competence in the planning and management of water and land areas (except for the PORTODIMARE project).
6. Taking account of the international and cross-border aspects in the development of maritime spatial planning and the draft plan in particular.
7. Continuous upgrading of information infrastructure and contribution to the development of databases (focal points in the SHAPE, SUPREME, ADRIPLAN, and PORTODIMARE projects).

In addition to the above-mentioned elements, the content development of these projects or their complementarity is also visible, i.e., SHAPE → ADRIPLAN; SHAPE + ADRIPLAN → SUPREME; SHAPE + ADRIPLAN + SUPREME → PORTODIMARE. With the gradual development and upgrading of the same elements, the legitimacy and applicability of the projects was increasing from project to project. Certain challenges were thus identified several times, which reflects in concrete results during the preparation of the draft plan. From the point of view of the project applicability in the preparation of the draft plan, the following three categories were defined:

1. Fully applicable elements, i.e., fully developed and directly transferable elements (e.g., consensus on ICZM coastal zone, coordinated with local communities; prioritizing of sectoral fields as set out in the MSP Directive; formulated protected areas and protection regimes, the Geoportal, and tools; and international starting points).
2. Largely applicable elements, i.e., developed methodology and the content structure for the draft plan preparation, harmonized cross-sectoral interests, identified stakeholders, defined administrative aspects, and cross-border coordination of the impacts of certain activities at sea.
3. Conditionally applicable or indirectly transferred elements, i.e., identified and already consensual local interests, specific data, sectoral expert bases, and spatial solutions.

According to all three categories, it can be concluded that the contribution of the projects is certainly multifaceted. The planning process was developing along with their implementation: therefore, the project elements considered are not only related to the content of the draft plan, but also have procedural and methodological significance (Table 14). Moreover, their financial contribution is also not negligible. Finally, fully applicable contents were financed or cofinanced through the projects (e.g., definition of the ICZM coastal zone, drawing of a plan for the Strunjan fishing port and the consequent adjustment of land use, etc.). The material prepared within these projects contributed also to the faster preparation of the draft plan.

## 4. Discussion

The research results indicate that longstanding and continuous implementation of the project activities in the fields of MSP promotion, identification of interests and conflicts, mobilization of stakeholders, creation of databases for the whole national sea and coast, coordination with international platforms, etc., contributed to the implementation of sustainable development principles and, within this framework, to the coherence of development and protection challenges. Through the project implementation process, the entire MSP process was gradually put in place, as well as individual methodological and content elements of the plan itself. In this sense, their contribution is certainly multifaceted. Along with the projects, a large base of knowledge, experience and international partnerships were formed and accumulated at the national level. Similar findings, as identified in the case of Slovenia, have been noted also in other studies to date [17,54]. Therefore, the significance of the research results is reflected in the comprehensive assessment of preliminary activities in the field of MSP development. This applies to the Northern Adriatic and the wider area. The present results may also provide suggestions to planners and

decision-makers in the process of preparation and implementation of similar projects not necessarily bound to the marine environment.

**Table 14.** Contribution of the projects to the development and preparation of the draft plan.

| Level of Contribution | Basic Elements | Detailed Elements |
| --- | --- | --- |
| Procedural Contribution | Establishment of MSP | Development of planning process stages<br>Mobilization of stakeholders and decision-makers<br>Implementation of MSP in legislation and practice<br>International aspect |
| Methodological Contribution | Phases of plan preparation<br>Working methods | Identification of interests (land-sea)<br>Identification of stakeholders and their mobilization<br>Balancing of interests<br>Development of databases and analytical tools |
| Structural Contribution | Design of project content<br>Concrete and partial results | Starting points and objectives of the plan<br>Sectoral fields<br>Plan of the maritime uses and activities<br>Design for the ICZM coastal zone<br>Plan implementation measures |
| Financial Contribution | Expert bases<br>Spatial design solutions | Services and products (partially or fully applicable in the draft plan) already paid for |
| Contribution in Terms of Time | Expert bases<br>Spatial design solutions | Services and products (partially or fully applicable in the draft plan) already produced |

The present research supports the statement [17,53] that the establishment of international projects is the right framework for further steps in the field of MSP also in the case of Slovenia. In this field, Slovenia does not have sufficient experience and specific expertise, which can be compensated by this kind of cooperation. The results show that the projects enable focusing on thematic fields, which are especially relevant for Slovenia. In particular, this includes the planning and management of the narrow coastal zone and partnership cooperation in the strategic management of the wider Northern Adriatic region, which has impacts on the Slovenian sea. Furthermore, Slovenia can also promote some of its good practices, including especially the effective system of protected areas management, developed and verified approach for the feasible implementation of the ICZM Protocol, and the effective system of cooperation between institutions at the local and national levels. Even after the implementation of the plans in all three countries of the Northern Adriatic, the plans will later be cyclically amended and supplemented according to the environmental challenges and sectoral interests. It is in this direction that we see the existence of such projects as a necessary cooperation that ensures active participation of all countries in the joint management of the sensitive marine environment.

## 5. Conclusions

In the area of the Northern Adriatic, several international projects have taken place in recent decades, which have significantly contributed to the development of MSP and the preparation of the first generation of maritime spatial plans. It was found that there were seven key projects implemented in Slovenia, cofinanced from the EU and UNEP-MAP programs, namely the CAMP Slovenia, PLANCOAST, SHAPE, ADRIPLAN, ADRIATIC+, SUPREME, and PORTODIMARE projects. Unlike in other countries, in Slovenia the projects were always applied to the entire national sea and the coastal zone but focused on project-specific topics. This feature is also an advantage since all projects were implemented within the scope of the formal maritime spatial plan and in one case even beyond.

The project elements that significantly contributed to the drafting of the first plan include: preliminary harmonization with international commitments and professional standards, comprehensively defined sectoral challenges and established cooperation between their representatives, established links between spatial planning bodies at local and

national levels, databases (platforms) and analytical tools, and the informed and involved professional and general public.

In determining and assessing the importance of these projects in the preparation of the first draft plan, it was found that they contributed at different levels. Through their implementation and results, and the materials prepared during their implementation, the projects contributed at: (a) the process level—support for the MSP development, (b) the methodological level—phasing and methodology of plan preparation, and (c) the structural level—elaboration of a detailed content of the plan. All the above also led to financial relief and time saving in the preparation of the first draft plan.

In the future, it would be reasonable to focus even more on the research of the relevance and results of international projects in the field of MSP on the role and activities of the participating institutions. In individual countries, the institutions face very different challenges in the management of marine environment, and through their participation, they are becoming a kind of "para-formal" stakeholders at the international level. The experience in the development and implementation of MSP is of utmost importance for all participants in the MSP process also outside the frameworks of such projects.

Another course of research can be aimed at the first generation of plans. In this context, there is a need to: (a) research the response to the plan (general and professional public, economy, public administration, and politics), and (b) evaluate their actual impact in the preparation of starting points for possible amendments to the first plan or starting points for all subsequent plans. A narrower field of research, which has not yet been sufficiently treated, is an effective monitoring system and appropriate indicators for monitoring the status of the marine environment. This is an instrument for implementing the plan towards achieving the objectives of the good status of the marine environment in accordance with the Marine Strategy Framework Directive [55].

Any research in the above fields can contribute in a specific way to the promotion and dissemination of good practices and is therefore in the wider public interest in both national and international terms.

**Author Contributions:** Conceptualization, G.Č. and B.R.; Data curation, V.U.; Formal analysis V.U. and G.Č.; Investigation, S.M. and V.U.; Methodology, G.Č., B.R., and S.M.; Project administration, V.U. and S.M.; Supervision G.Č.; Visualization V.U. and B.R.; Writing—Original draft, G.Č., B.R., and S.M.; Writing—Review and editing, G.Č. All authors have read and agreed to the published version of the manuscript.

**Funding:** This research was partly funded by Slovenian Research Agency, by the Research Program "Geoinformation infrastructure and sustainable spatial development of Slovenia", no. P2-0227.

**Institutional Review Board Statement:** Not applicable.

**Informed Consent Statement:** Not applicable.

**Data Availability Statement:** The data presented in this study are available on request from the corresponding author.

**Conflicts of Interest:** The authors declare no conflict of interest.

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
