# Peer review of "Contribution of International Projects to the Development of Maritime Spatial Planning Structural Elements in the Northern Adriatic: The Experience of Slovenia"

_water, doi:10.3390/w13060754_

Round 1
Reviewer 1 Report
This is an interesting communication paper dealing on the implementation of the MSP directive and elaboration of maritime spatial plans. It reviews and analyses seven projects and identifies key elements in relation to the MSP directive. It also highlights the importance of international cooperation to assure the success of the implementation and to allow a joint management of the marine environment.
However, the paper needs minor revisions mostly regarding the "low" description of the results (in particular the tables and figures) that are:
- Table 4 is not strongly useful, some information is also indicated in figure 1. Therefore, I suggest to delete table 4 by integrating relevant information in table 3 (extension of the surface covered by the project and/or percentage considered). Figure 1 should be better explained.
- Maybe table 6 can also be deleted because the information reported in this table can be directly put in the text.
- The percentage reported in table 8 are not clear, please provide further information on how the percentage was calculated (147% of o.ther stakeholders?)
- Are figures 2a and 2b useful? Please provide further information and detail on the figures. The figures appear very similar, what are the differences?
- Maybe the information reported in lines 427 to 431 should be placed at the beginning.
- Table 5, 6, 9 and 15 are not cited in the text
English should be checked.
Author Response
Please see the attachement

Reviewer 2 Report
Why is this research relevant on the global level, how can this be a contribution for other cases in MSP? Please specified this better in the Introduction.
Please remove bullets from the Tables 1,2 and 15. Check spelling in Table 14.
Please separate Discussion and Conclusion section. Discussion should give the comparison with previous studies while Conclusion must contain main conclusions based on the achieved results.
Future directions are also necessary.
Author Response
Please see the attachement

Reviewer 3 Report
This is a beautifully presented and useful paper, very original in its approach, aiming to show how the existence of a series of international MSP related (transnational or territorial cooperation) projects in the Adriatic ( Eastern mediterranean ) sea has efficiently influenced the Maritime Spatial Planning process on a national level, particularly taking as an example Slovenia. There is a lot of useful information presented and analysed in the paper that deserves publication in its present form.
I would simply add some remarks/ideas that authors are not obliged to follow but could inspire further research.
- I would like to see how all these projects interact or not with the Adriatic-Ioanian Macro-Regional strategy and its Blue Growth pillar (EUSAIR)
- Another idea is how these projects have contributed or not to the consultation procedures of the draft MSPlan of Slovenia.
- A third comment is that Slovenia is the most advanced country in the Eastern Mediterranean as far as the implementation of a Maritime Spatial Plan is concerned. I would urge the authors to examine if the international projects presented in their research have finally contributed to the acceleration of the national MSP procedures by transferring relevant knowledge etc.
Author Response
Please see the attachement

Round 2
Reviewer 2 Report
Authors included all reviewer's comments.